# The Mouse CircGHR Regulates Proliferation, Differentiation and Apoptosis of Hepatocytes and Myoblasts

**DOI:** 10.3390/genes14061207

**Published:** 2023-05-31

**Authors:** Weilu Zhang, Shudai Lin, Zhenhai Jiao, Lilong An, Tingting Xie, Jiang Wu, Li Zhang

**Affiliations:** 1College of Coastal Agricultural Sciences, Guangdong Ocean University, Zhanjiang 524088, China; zhangweilu1211@foxmail.com (W.Z.); linsd89sylvia@163.com (S.L.);; 2Experimental Teaching Center, Guangdong Ocean University, Zhanjiang 524088, China

**Keywords:** mouse, circGHR, NCTC1469, C2C12, cell proliferation

## Abstract

The anterior pituitary gland of animals secretes growth hormone (GH) to bind to the growth hormone receptor (GHR) on the liver cell membrane through the blood circulation, thereby promoting the downstream gene insulin-like growth factor-1 (IGF1) expression, which is the canonical GH–GHR–IGF1 signaling pathway. Therefore, the amount of GHR and the integrity of its structure will affect animal growth and development. In the previous study, we found that the mouse *GHR* gene can transcribe a circular transcript named circGHR. Our group cloned the full-length of the mouse circGHR and analyzed its spatiotemporal expression profile. In this study, we further predicted the open reading frame of circGHR with bioinformatics, subsequently constructed a Flag-tagged protein vector and preliminarily verified its coding potential with western blot. Additionally, we found that circGHR could inhibit the proliferation of NCTC469 cells and has a tendency to inhibit cell apoptosis, while for C2C12 cells, it showed a tendency to inhibit cell proliferation and promote its differentiation. Overall, these results suggested that the mouse circGHR had the potential to encode proteins and affect cell proliferation, differentiation and apoptosis.

## 1. Introduction

Circular RNA (circRNA) is a covalently closed circular structure lacking a 5′ cap or 3′ tail and has been classified as non-coding RNA (ncRNA) since it was discovered and recognized. In 1976, four highly purified viroids were identified as circular RNAs with special properties, such as high thermal stability, cooperativity and self-complementarity [1]. Surprisingly, it was shown that circRNAs are abundant, conserved and highly expressed in animals [2,3]. The closed-loop structure of circRNA provides it with higher stability, making it less susceptible to degradation by RNase R [4,5]. Subsequent studies found that circRNA can not only regulate transcription, act as a miRNA molecular sponge and bind proteins, but it can also translate proteins by itself [6,7,8]. Until now, more and more research has revealed that circRNA plays important roles in humans, rabbits, cows, monkeys, sheep, pigs and other organisms [9,10,11]. Numerous studies have shown that some circRNAs are expressed and play important roles in mouse liver. For instance, a total of 93 dysregulated circRNAs were observed in mice with nonalcoholic fatty liver disease [12]. What is more, it was displayed that mm9_ circ_018725 could inhibit hepatocyte apoptosis induced by EtOH in vitro [13]. In addition, research has reported that circRNAs can regulate skeletal muscle development in mice. For example, circHIPK3 can promote C2C12 cell proliferation and differentiation through the miR-7/TCF12 pathway [14]. Moreover, circARID1A may regulate mouse skeletal muscle cell development and regeneration by functioning as a sponge of miR-6368 [15].

It is well known that the neuroendocrine growth axis “hypothalamus–pituitary–growth hormone–target organ” regulates animal development. The growth hormone (GH) secreted by the anterior pituitary gland combines with the growth hormone binding protein (GHBP) and enters into various tissues through the blood circulation. Then, GH dissociates from GH–GHBP and binds with the GHR on the cell membrane of the target organ and, through the JAK–STAT pathway, initiates intracellular signal transduction and the expression of insulin-like growth factors (IGFs) [16,17]. Subsequently, IGFs affect skeletal muscle development by regulating myoblast or satellite cell differentiation and fusion [18,19,20]. The GH–GHR–IGF1 signaling pathway is involved in muscle growth and tissue development, which is of great significance for the growth and differentiation of animal skeletal muscle cells [21].

The pre-RNA of the GHR gene can be back spliced to form a circular transcript (circGHR) in human, mouse and chicken [16]. We previously performed high-throughput RNA sequencing of chicken liver and muscle, finding that chicken circGHR could promote the proliferation of primary myoblasts [22]. In addition, we found that mouse circGHR was formed by backward splicing of exons 2 and 8 of the *GHR* gene. The spatiotemporal expression pattern revealed that mouse *GHR* mRNA and circGHR were widely present in various tissues. Among them, circGHR was highly expressed in mouse liver, spleen, lung and kidney but lowly expressed in muscle tissues. However, the roles of mouse circGHR in liver and muscle development were still unclear. Therefore, in order to explore its function in hepatocytes and myoblasts, we cloned mouse circGHR in a Flag-tag vector, verified its protein coding potential using the western blot method and then conducted the cell proliferation, differentiation and apoptosis analysis.

## 2. Materials and Methods

### 2.1. The Open Reading Frame (ORF) of Mouse circGHR

For the bioinformatics analysis of the ORF of the mouse circGHR, two times of its full-length sequences (a total of 1640 nt) were copied into the online website Open Reading Frame Finder of the National Center for Biotechnology Information (NCBI) (https://www.ncbi.nlm.nih.gov/orffinder/ (accessed on 1 March 2021) to predict its potential ORF.

### 2.2. RNA Extraction and Complementary DNA (cDNA) Synthesis

We used the HiPure Universal RNA Mini Kit (Magen, Guangzhou, Chian, R4130) to extract total RNA from mouse liver tissues according to the kit’s instructions. Then, the total RNA of the tissue was reverse transcribed into cDNA using the reverse transcription reaction components containing 2 μL RNA, 1 μL Anchored Oligo (dT)_18_ Primer or Random Primer, 10 μL of 2 × TS Reaction Mix, 1 μL TransScript^®^ RT/RI Enzyme Mix, 1 μL gDNA Remover and 4 μL RNase-free Water, and performed with the polymerase chain reaction (PCR) program, which was designed according to the instructions.

### 2.3. Vector Construction

The full-length sequence of circGHR was amplified via a 20 μL reaction system including 2 μL cDNA, 0.5 μL Forward primer, 0.5 μL Reverse Primer, 10 μL of 2 × EasyTaq^®^ PCR Super Mix and 7 μL double-distilled water, and the following reaction procedure was followed: hold at 95 °C for 30 s for pre-denaturation, 35 cycles of denaturation at 95 °C for 15 s, annealing at 57 °C for 30 s, and extension at 72 °C for 30 s followed by extension at 72 °C for 5 min. Then, the whole sequence of circGHR was cloned into the pCD2.1-ciR vector (Geneseed Biotech, Guangzhou, China) to obtain the plasmid pCD2.1-circGHR using the restriction enzymes KpnI and BamHI (Takara Bio, Beijing, China, 1010S, 1068S) according to the reagent instructions. Then, the Flag tags’ sequence was inserted before the stop codon “UGA” of pCD2.1-circGHR to construct the plasmid pCD2.1-circGHR-Flag.

### 2.4. Cell Culture and Treatment

Mouse hepatocytes NCTC1469 and myoblasts C2C12 were cultured in 10% growth medium (GM) (DMEM, Gibco, Thermo Fisher Scientific, Waltham, MA, USA, 11995065 with 10% fetal bovine serum (Gibco) and 0.2% penicillin/streptomycin (Invitrogen, Carlsbad, CA, USA, 15140122)) and placed in a 37 °C, 5% CO_2_ incubator. GM was replaced every two days, and the differentiation medium (DM) (DMEM + 2% pregnant horse serum (Gibco, 16050122)) was used to induce C2C12 differentiation.

Mouse hepatocyte NCTC1469 cells were transfected with the plasmids using Lipofectamine 3000 reagent (Invitrogen, L3000008) when 90% confluence was reached following the manual. C2C12 cells were transfected at 50% confluence and cultured in GM for investigating cell proliferation. When the C2C12 cells were at 100% confluence, DM was used for differentiation studies.

### 2.5. Quantitative Real-Time PCR (qRT-PCR)

We isolated total cellular RNA using TRIzol reagent (Magen, Guangzhou, China, R4130-02) following the manufacturer’s instructions. Further, cDNA Synthesis SuperMix (TransGen Biotech, Beijing, China, AT311) was used to produce the cDNA according to the manufacturer’s protocol. The qRT-PCR was analyzed using SYBR Green qPCR Mix (DONGSHENG BIOTECH, P2091a-P2092a-P2093a) with gene specific primers, while the *GAPDH* gene was used as the internal control. The 20 μL qRT-PCR component of each reaction contained 1 μL cDNA, 0.4 μL Forward primer, 0.4 μL Reverse Primer, 10 μL of 2 × EasyTaq^®^ PCR Super Mix and 8.2 μL double-distilled water. Then, we performed the following qPCR program: hold at 95 °C for 30 s for pre-denaturation, 40 cycles of denaturation at 95 °C for 10 s and annealing and extension at 60 °C for 30 s followed by collecting the melt curve. The primers used in this study are listed in Table 1.

### 2.6. Cell Counting Kit 8 (CCK-8) and EdU Assay

Cells were seeded into 96-well plates and transfected with pCD2.1-circGHR and pCD2.1-ciR vectors. Cell proliferation was detected from 12 to 72 h using the Cell Counting Kit-8 kit (TransGen Biotech, Beijing, China, FC101) by adding 10 μL per well of CCK-8 solution and culturing for 1 h in a 37 °C, 5% CO_2_ incubator and then measuring with a Microplate Reader (Thermo Fisher Scientific) at 450 nm.

For the EdU assay, a total of 5 × 10^5^ cells/well were seeded in 12-well plates and transfected with the above two vectors using Lipofectamine 3000 reagent (Invitrogen, L3000008) according to the operating manual. After 48 h, the Cell-Light EdU Apollo567 In Vitro Kits (RiboBio, Guangzhou, China, R11053.9) were used; the cells were incubated with 50 µM EdU reagent at 37 °C for 2 h, and the cell nuclei were counter-stained with Hoechst 33342 for 30 min, which was performed according to the manufacturer’s protocol. Subsequently, a fluorescence microscope was used to capture three randomly selected fields per well. Cells were seeded in 12-well plates and transfected with the above two vectors. We referenced the introduction and the literature [23] to count the Edu-stained cell number with Image-Pro Plus software.

### 2.7. Western Blot Assay

Cells were seeded into 6-well plates and transfected with pCD2.1-circGHR-flag vectors. After culturing for 48 h in DM, the total protein of the cells was extracted by the lysate (Beyotime, Shanghai, China, P0013B, ST506) (RIPA lysate and PMSF were prepared in a ratio of 10:1). We used the bicinchoninic acid (BCA) protein assay kit (Beyotime, Shanghai, China, P0010S) to detect the protein concentration. The subsequent SDS-PAGE, polyvinylidene fluoride (PVDF) membrane was transferred, and the hybrid was referenced in the literature [23]. The primary detection antibodies (Beyotime, 1:1000), including anti-Flag tag antibody (Beyotime, AF519), anti-cyclin D1 (CCND1) (Beyotime, AF1183) rabbit monoclonal antibody, anti-cyclin-dependent kinase 2 (Cdk2) rabbit monoclonal antibody (Beyotime, AF1063), anti-proliferating cell nuclear antigen (PCNA) mouse monoclonal antibody (Beyotime, AF0261) and anti-β-actin antibody (Beyotime, AA128), were used. The PVDF membrane was washed 3 times with western wash buffer (Beyotime, P0023C6) followed by incubation for 2 h at room temperature with the horseradish peroxidase (HRP)-labeled goat anti-mouse IgG(H+L) (Beyotime, A0216) or (HRP)-labeled goat anti-rabbit IgG(H + L) (Beyotime; 1:1000) (Beyotime, A0208). Finally, antibody reactive bands were detected using the OmniECL Femto light chemiluminescence kit (Epizyme Biomedical Technology Co., Ltd., Shanghai, China, SQ201). The relative expression of protein was measured with Image J software.

### 2.8. Flow Cytometric Cell Apoptosis Analysis

A total of 5 × 10^5^ NCTC1469 cells per well were cultured into 12-well plates and transfected with plasmids when they were completely attached. After being transfected with the plasmids for 48 h, the cells were treated with the Annexin V-FITC/PI apoptosis detection kit (Vazyme, Nanjing, China, A211), and the percentage of NCTC1469 apoptosis was detected with flow cytometry.

### 2.9. Immunofluorescence Assay

To analyze the roles of mouse circGHR on cell differentiation, we performed an immunofluorescence assay according to the reference [23]. Simply, C2C12 cells were seeded and cultured in DM. After being transfected with the plasmids for 12 h, 48 h and 96 h, the cells were fixed for 30 min and then blocked with 5% goat serum (Beyotime, C0265) for 30 min. After incubation with anti-MyHC (Proteintech Group, Wuhan, China, 1:500, 22280-1-AP) at 4 °C for 12 h, CY3 Conjugated AffiniPure Goat Anti-rabbit (Boster, Wuhan, China, 1:1000, BA1032) was added followed by incubation at 37 °C for 1 h. The cell nuclei were stained with DAPI (Sangon Biotech, Shanghai, China, 1:50, E607303-0002). Three images were randomly selected to calculate the area of myotubes with Image-Pro Plus software.

### 2.10. Statistical Analysis

Data were listed as the mean ± standard error of the mean (SEM). The relative expression of the genes was analyzed with the 2^–ΔΔCT^ method according to the reference [24]. Differences between the means were analyzed with the unpaired Student’s *t*-test. All analysis was performed using SPSS 20.0 software (SPSS Statistics IBM, New York, NY, USA). *p* < 0.05 was considered to indicate statistical significance.

## 3. Results

### 3.1. Mouse circGHR Encodes Novel Polypeptide circGHR-295aa

The bioinformatics analysis of the ORF structure of circGHR showed that the ORF structure of this circle molecule encodes from the 10th base of exon 2 (ATG), surrounds the whole circRNA sequence, spans the circRNA back-splice junction and terminates at the 77th base of exon 2 (TGA), indicating that the mouse circGHR has an open reading frame (ORF) of 888 bp in length, which could encode a polypeptide of 295 aa (Figure 1A). The initiation codon (AUG) of the ORF started from the 10th base of exon 2, and the ORF terminated at the 77th base (UGA) of exon 2. The ORF crossed the junction position and wrapped around the circular RNA.

In order to verify the coding potential of the mouse circGHR, we constructed a Flag-tagged protein carrier vector named pCD2.1-circGHR-Flag (Figure 1B) and transfected it into C2C12 cells for 48 h (Appendix A); then, we collected the cells and detected the Flag-tagged protein with western blot. The ORF of circGHR was 888 nt in length, which could encode a polypeptide with a molecular weight of 33.6 kDa. The Flag tag was a marker protein composed of eight aa; therefore, the molecular weight of the pCD2.1-circGHR-Flag fusion protein was 36.6 kDa. As shown in Figure 1C, the fusion protein was not detected in the negative group by the flag antibody, while the product was detected in the pCD2.1-circGHR-Flag transfected group, suggesting that the mouse circGHR can encode a polypeptide.

### 3.2. CircGHR Inhibits the Proliferation of Mouse NTCT1469

In this study, we conducted CCK-8, qRT-PCR and western blot experiments to analyze the effect of circGHR on the proliferation of NCTC1469 cells. After transfecting the pCD2.1-circGHR into NCTC1469 cells, the CCK-8 results indicated that the cell viability decreased at six time points, but it was higher in the pCD2.1-circGHR group than that of the control group (*p* > 0.05) (Figure 2A). As shown in Figure 2B, circGHR in the overexpression vector group was significantly higher than that in the control group at 48 h (*p* < 0.01); in addition, the expression level of *GHR* mRNA was significantly higher than that in the empty vector group (*p* < 0.05), while the three proliferation marker genes in the control group were higher than those in the pCD2.1-circGHR group. Among them, the *CCND1* gene was significantly higher than that of the empty vector group (*p* < 0.05). From the results of western blot, the CCND1 protein expression levels were consistent with the qRT-PCR results (Figure 2C, Appendix A).

### 3.3. CircGHR Tends to Inhibit NCTC1469 Apoptosis

To investigate the effect of circGHR on the apoptosis of NCTC1469 cells, three marker genes were detected with qRT-PCR analysis after the cells were transfected with pCD2.1-ciR and pCD2.1-circGHR. We found that *Fas* and *Bax* were significantly lower than the control (*p* < 0.05, Appendix A). The results of flow cytometry showed that the apoptosis rate of the circGHR cells was lower than that of the control, and the trend was consistent with the qRT-PCR analysis, which indicated that circGHR had a tendency to inhibit the apoptosis of NCTC1469 cells.

### 3.4. CircGHR Has No Effects on the Proliferation of C2C12

The cell survival rate of the circGHR group was higher than that of the control group at six different time points. In addition, the expression level of circGHR was significantly higher in the C2C12 cells than in the control group (*p* < 0.01, Appendix A), while the *GHR* mRNA showed a lower trend than the control group (*p* < 0.05). However, there was no significant differences in the expression levels of the three proliferative marker genes between the two groups. From the results of the Edu assay, we found that the proliferation rate of the C2C12 cells did not show any obvious change after 48 h overexpression of circGHR.

### 3.5. CircGHR Promotes C2C12 Differentiation

In order to explore the effect of the mouse circGHR on cell differentiation, we used qRT-PCR to detect the changes in differentiation marker genes. After the cells were induced by 2% pregnant horse serum, qRT-PCR results showed that circGHR was significantly higher than that in the control group at 48 h (*p* < 0.01), and the *GHR* did not show significant differences between these two groups (*p* > 0.05). In addition, the three differentiation marker genes (*MyoG*, *MyHC* and *MyMK*) in the circGHR group were all a little higher than those in the control group (*p* < 0.05) (Figure 3A). Notably, the MyHC immunofluorescence assay results exhibited a significant increase in the circGHR group compared with the control (Figure 3B) with a trend of having increased, prolonged and brightened myotubes at 12, 48 and 96 h (Figure 3C–E). This result was consistent with the trend of qRT-PCR, which indicated that mouse circGHR could promote C2C12 differentiation.

## 4. Discussion

Recently, more and more studies have demonstrated that many circRNAs can be translated and function in myogenesis or perform other cellular roles. In 2017, it was first discovered that the circular RNA molecule circMbl in Drosophila can translate proteins and perform biological functions [25]. Subsequently, circular RNA derived from the zinc finger protein 609 gene (termed circZNF609) was found in humans and mice [3]. Another research study found that the protein encoded by the chicken ring molecule circFAM188B may regulate the proliferation and differentiation of skeletal muscle cells, participating in muscle development [23]. In human disease research, it has also been found that circRNAs that can translate proteins play important roles in cancer cells, such as circSHPRH, circAKT3 [26] and circNSUN2 [27,28]. With the continuous deepening of research on circRNA, its ability to encode proteins has attracted much attention. Several databases and websites have emerged to collect RNA with the ability to encode proteins, such as http://www.jianglab.cn/ncEP/ (accessed on 14 December 2020). However, only a small amount of circRNA data is included, and a large amount is still unknown. Especially, there are few research reports on the mechanism of circRNA regulating skeletal muscle development by encoding proteins. Currently, only circZNF609 [3] and circFAM188B [23] are found to regulate muscle development through encoding proteins.

The coding potential of the circular RNA can be verified using different methods. For example, we can screen out circRNAs that bind to ribosomes using ribosomal imprinting and deep-sequencing technology [20,29], or we can insert the tag sequence of Flag, His, MBP or GST before the stop codon of ORF, which could be detected with western blot or immunohistochemistry using the specific antibodies or with immunofluorescence probes [30]. Previously, researchers found that many circRNAs have protein-coding capacity, such as circPPP1R12A [31], circFBXW7 [32], circFAM188B [23] and circZNF609 [3]. Chicken circFAM188B could encode a novel protein (circFAM188B-103aa), promoting cell proliferation and inhibiting the differentiation of skeletal muscle satellite cells [23]. In the current study, we determined that the mouse circGHR encoded a 295 aa peptide. However, its function in animal development needs further study.

Research has shown that there are four mechanisms for circRNA to translate proteins: (i) Ribosome entry site (IRES) mediated translation [33]; (ii) roll ring amplification translation (DNA rolling circle amplification (RCA) is a way for bacteriophages to self-replicate after infecting viruses, allowing for infinite single-stranded amplification of circular DNA molecules, and some circRNAs, due to the absence of termination codons, have infinite open reading and can roll around to translate proteins such as RCA, producing a large number of protein products) [34]; (iii) translation mediated by translation activation elements in an untranslated region (UTR) (CircRNAs that contain gene UTR sequences could drive circRNA to translate protein by recruiting ribosomes) [25]; and (iv) translation mediated by m6A modification [35]. However, the mechanism underlying the mouse circGHR encoding protein is still unclear, which needs further research.

According to the “hypothalamus–pituitary–growth hormone–target organ” endocrine growth axis, GH first reaches the liver through the blood circulation. Studies have found that circRNAs play important roles in the liver. The circRNA_0046367 could change the process of liver steatosis [36], while the circRNA-0067835 participates in the occurrence of hepatic stellate cell fibrosis [37]. Many circRNAs were screened out through high-throughput sequencing of RNA from patients with chronic hepatitis [38]. The circRNA may be involved in the process of liver regeneration, repair and occurrence of hepatocellular carcinoma by forming a circRNA–miRNA–mRNA regulatory network, such as the circRNA-5692 that could inhibit the development of liver cancer through the circRNA-5692/miR-328-5p/DA B2IP pathway [39]. What is more, circ-0091579 and miR-940 regulate the expression of TACR1, which affects HCC development [40]. These reports indicated that circRNAs had a significant impact on the liver and hepatocytes. In this study, we found that circGHR could inhibit the proliferation and apoptosis of mouse NCTC149 cells. The qRT-PCR results showed that the expression of *GHR* mRNA at 48 h was significantly higher than that of the control group after being transfected with pCD2.1-circGHR (*p* < 0.05), which hinted that the level of circGHR might be correlated to the mRNA of the *GHR* gene, hence affecting NCTC1469 cell proliferation.

When IGF1 reaches skeletal muscle, it also plays key roles in its growth and development. There have been many reports that a large number of circRNAs function in the skeletal muscle of humans and animals, and their expression is spatiotemporally specific [30]. One study sequenced rhesus monkey skeletal muscle, detected 12,007 circRNAs and found that the expression of five circRNAs decreased with the age of the rhesus monkeys [10], while some circRNAs decreased significantly with the progress of growth and development [41]. Similarly, circRNAs also played pivotal roles in myoblast proliferation, differentiation and apoptosis. It was discovered that circLMO7 can inhibit myoblast differentiation, promote myoblast proliferation and avoid apoptosis of bovine primary myoblasts [42]. In addition, many circRNAs, such as circSVIL [43], circFGFR4 [44], circFUT10 [45] and circCDR1as [46], could promote myoblast proliferation and differentiation through acting as a sponge of miRNAs. Besides, the circSamd can bind to proteins to promote myoblast differentiation [47]. In this study, circGHR had no effects on the proliferation of C2C12 cells. Mechanically, the *GHR* mRNA in the circGHR-overexpression group was slightly lower than that in the control group, which suggests this circGHR may have an antagonistic effect on the transcription of linear mRNA, therefore, resulting in the C2C12 cell proliferation inhibition. MyHC and Myomaker genes functioned in myotubes formation processes, including muscle cell migration, recognition, adhesion and fusion. Myomaker can independently regulate different stages of myoblast fusion, mainly acting on the semi-fusion of membranes [48], while the MyHC gene is a terminal differentiation gene of myoblasts and acts from myocytes to myotubes [49]. In this study, mouse circGHR had a certain promoting effect on the process of terminal cell differentiation. Moreover, qRT-PCR and immunofluorescence results suggested that circGHR may have a promoting effect for the differentiation process of C2C12 cells through stimulating the expression of the MyHC gene.

## 5. Conclusions

In short, we preliminarily confirmed that the mouse circGHR was a circRNA with protein-coding potential, and it could promote cell proliferation, tended to inhibit apoptosis of NTCT1469 cells and had a tendency to function in proliferation and promote differentiation of myoblast C2C12 cells.

## Figures and Tables

**Figure 1 genes-14-01207-f001:**
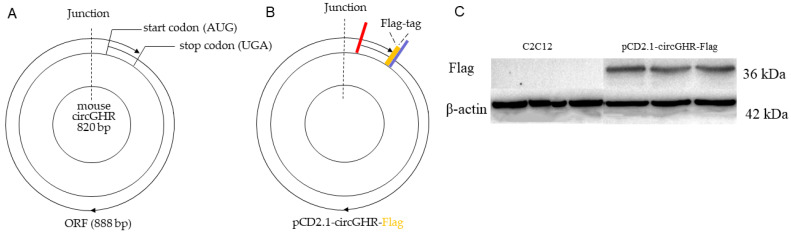
Schematic diagram and protein-coding capacity of the mouse circGHR. (**A**) The predicted coding sequence and its length of the mouse circGHR; (**B**) Schematic diagram of the pCD2.1-circGHR-Flag vector; **—**: **start codon**, **—**: **stop codon**, **—**: **Flag-tag**; (**C**) The expression level of pCD2.1-circGHR-Flag detected by flag antibody after transfecting into C2C12 cells.

**Figure 2 genes-14-01207-f002:**
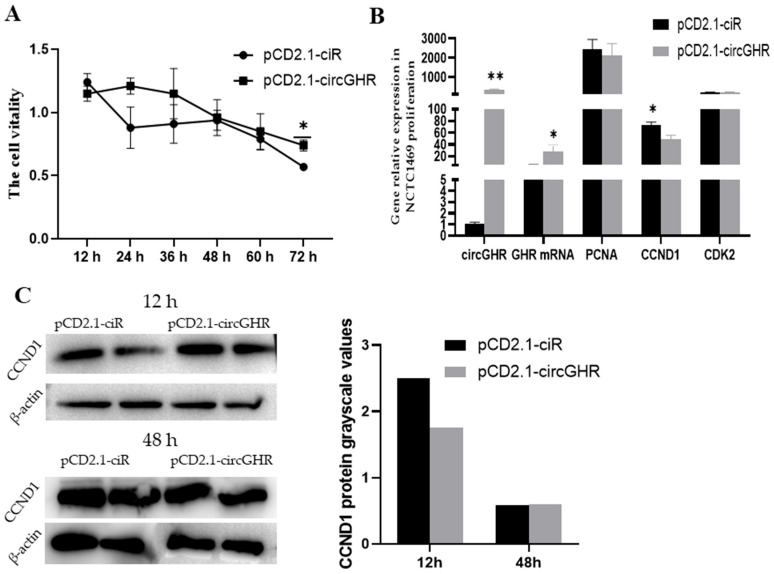
Cell viability and the expression levels of 3 proliferation marker genes after transfecting pCD2.1-circGHR into mouse NTCT1469 cells. (**A**) CCK-8 results of the cells being transfected with pCD2.1-ciR and pCD2.1-circGHR; (**B**) The level of circGHR, *GHR* mRNA, *CCND1*, *CDK2* and *PCNA* after transfection with pCD2.1-ciR and pCD2.1-circGHR for 48 h; (**C**) CCND1 protein detected with western blot after the cells were transfected with pCD2.1-ciR and pCD2.1-circGHR. The data are represented as the mean ± SEM; **: *p* < 0.01, *: *p* < 0.05.

**Figure 3 genes-14-01207-f003:**
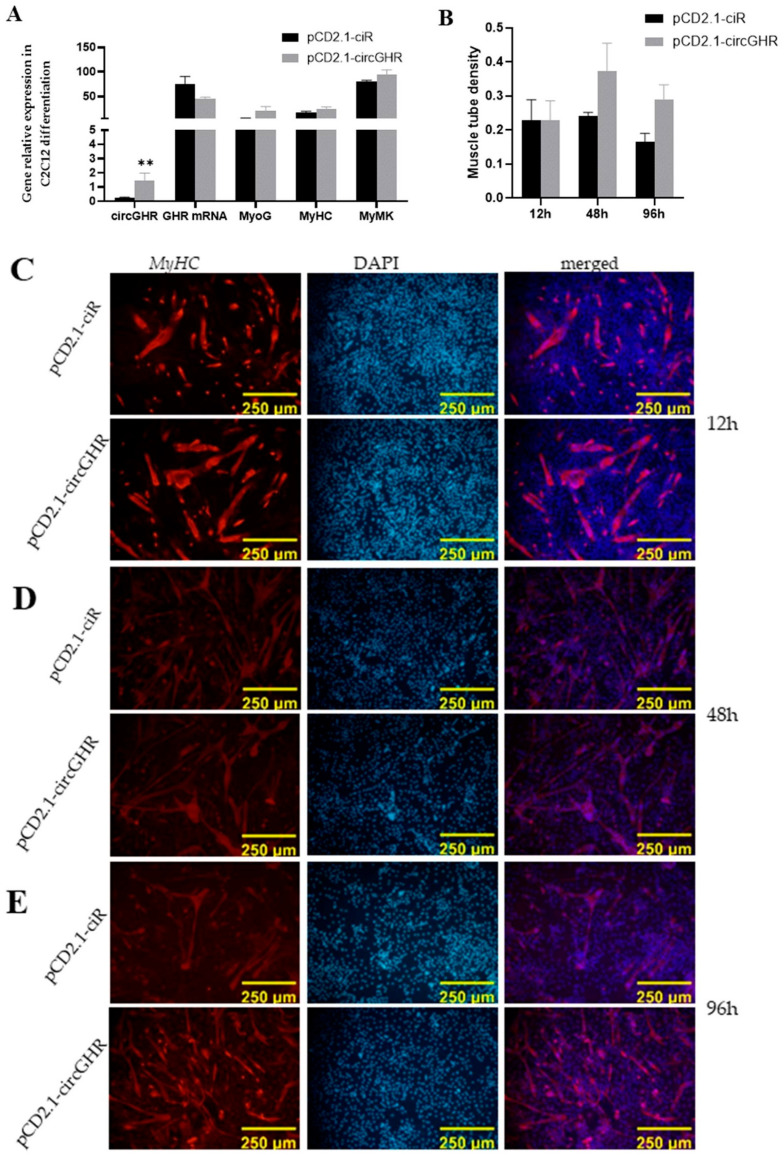
Effect of circGHR on the differentiation of C2C12 cells. (**A**) The expression level of circGHR, *GHR* mRNA, *MyoG*, *MyHC* and *MyMK* after transfection with pCD2.1-ciR and pCD2.1-circGHR for 48 h; (**B**) The muscle tube density of the C2C12 cells detected with MyHC immunofluorescence at 12, 48 and 96 h after transfection with pCD2.1-ciR and pCD2.1-circGHR; Immunofluorescence results of MyHC at 12 h (**C**), 48 h (**D**) and 96 h (**E**) after transfection. The data are represented as the mean ± SEM; **: *p* < 0.01.

**Table 1 genes-14-01207-t001:** Primer sequences for gene quantitative RT-PCR.

Genes	Primer Sequence (5′ to 3′)	Annealing Temp (°C)	Product Length (bp)
*circGHR-full*	F: GTCTCAGGTATGGATCTTTGTCAGG R: CTTCTTCACATGCTTCCAATATGTTC	57	820
*circGHR-DL*	F: GGGATTCGTGGAGACATCCAA R: GACTGCCAGTGCCAAGGTTA	59	343
*circGHR-Flag-DL*	F: AACCTGATCCACCCATTGGC R: ATCTCACCCGCACTTCATGT	58	252
*PCNA*	F: ACCTCACCAGCATGTCCAAAA R: GGATTCCAAGTTGCTCCACATC	60	174
*CCND1*	F: AAAATGCCAGAGGCGGATGA R: GAAAGTGCGTTGTGCGGTAG	60	199
*CDK2*	F: GCCATTCTCACCGTGTCCTT R: GGACTCCAAAGGCTCTTGCT	60	111
*BAX*	F: TGAAGACAGGGGCCTTTTTG R: AATTCGCCGGAGACACTCG	58	140
*Fas*	F: GAAAGTCCAGCTGCTCCTGT R: ACACCAGGAGTTGCCAATGT	59	290
*MyoG*	F: GGCTGTCCTGATGTCCAGAAA R: CCAGAGGCTTTGGAACCGGATA	58	396
*MyHC*	F: CAAGTCATCGGTGTTTGTGG R: TGTCGTACTTGGGCGGGTTC	58	158
*MyMK*	F: CCTGCTGTCTCTCCCAAGGT R: GAACCAGTGGGTCCCTAAGC	59	133
*GAPDH*	F: AGGTTGTCTCCTGCGACTTCA R: TGGTCCAGGGTTTCTTACTCC	57	184

*PCNA*: proliferating cell nuclear antigen, *CCND1*: cyclin D1, *CDK2*: cyclin dependent kinase 2, *BAX*: BCL2-associated X, *MyoG*: myogenin, *MyHC*: myosin heavy chain, *MyMK*: myomaker, *GAPDH*: Glyceraldehyde 3-phosphate dehydrogenase.

## Data Availability

All data have been listed in the paper and the Appendix A. No other new data were created.

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
