# Peer review of "The Mouse CircGHR Regulates Proliferation, Differentiation and Apoptosis of Hepatocytes and Myoblasts"

_genes, 2023, doi:10.3390/genes14061207_

Round 1
Reviewer 1 Report
1) In the introduction, the authors indicated that the submitted manuscript is a followw up study after the cloning and expression analysis of chicken annd mouse circGHR. However, these cloning and expression analysis studies were cited in the manuscript. These initial studies should be added in the list of references.
2) It will be interesting to add (in Fig 1 or as separate figure) phase images of the myoblasts and hepatocyte cell lines at different time points following their transfection with circGHR as compared to control cells.
3) Add quantitative/densitometric analysis for the protein bands of Fig 2 and try to improve the quality of the blots (some bands are really bad).
4) The are many typos and grammer mistakes that need corrections.
Reviewer 2 Report
The manuscript presented deals with The Mouse CircGHR. Unfortunately, the manuscripts suffers several major weaknesses:
Materials are often poor described: It remains unclear which kits were used. No order-# etc. are mentioned, the manufacturer Geneseed Bio-tech, Guangzhou, China can´t be found in the web, according to their home-page Cell BioLabs, Shanghai, China obviously doesn´t offer restriction enzymes. The reader must have the opportunity to understand what was done. On the other hand, many space consuming details are described, this can be covered with reference to the manufacturer's instructions for use.
It remains unclear what the authors have published in their paper entitled “Cloning and expression of circular transcript of mouse growth hormone receptor gene” by Zhang WL et al Yi Chuan . 2021 Sep 20;43(9):890-900. doi: 10.16288/j.yczz.21-156. The title is very close to the manuscript presented, however, the 2021-paper is not referenced and not available to the reviewer since it is in Chinese language. Another publication of the authors´ group close to the topic of the manuscript is not referenced and discussed: Xu H et al. Front Genet. 2021 Feb 11;12:598575. doi: 10.3389/fgene.2021.598575.
Something with Table 1 is wrong (lines 87 to 116), obviously two segments of the manuscript were mixed up.
The manuscript is overloaded with Figures, altogether about 30 sub-figures are presented making it very difficult for the reader to understand what was done. The concluding message of the Figures remains often open, in many cases an example giving Figure would be much more helpful, e.g. in Fig. 2C or Fig. 5. Negative results take a lot of space, e.g. Chapter 3.4 as well as trends in Chapter 3.3. These data without significance can be mentioned in brief but don´t need these details.
Details of statistical calculations are not shown, the reviewer has major doubts how a statistical significance can be calculated with n=4.
Discussion: Lines 276 – 279 can be deleted, this should be known in general.
As mentioned by the authors, any functional (in vivo) studies are lacking but crucial to understand the role of circGHR. Their statement is the Conclusion-section is “…we preliminarily confirmed that the mouse circGHR was a circRNA with protein-coding potential…”. The reviewer thinks that this study needs more results to understand the role of circGHR and subsequently doesn´t qualify for publication in the current state.
Round 2
Reviewer 2 Report
The quality of the presentation of the data is much better now, especially by the shift of not significant data to a supplement. However, a not solvable issue is that no in vivo data are available. In any case, language and layout improvements are needed, e.g. double full stops in line 139.
Author Response
- Comment: The quality of the presentation of the data is much better now, especially by the shift of not significant data to a supplement. However, a not solvable issue is that no in vivo data are available. In any case, language and layout improvements are needed e.g. double full stops in line 139.
Response: Thanks again for your precious suggestion to improve our manuscript. We have revised the manuscript sentence by sentence guided by a native English professor of animal science. About the mechanism of action of circGHR in vivo, as well as the effect of the action is indeed a very important aspect. Yet, we have not begun these works. So, we will continue to explore it in depth in our subsequent studies under your encourage, many thanks for you.
We want to express our great appreciation to you the Editorial Board and Reviewers for their constructive comments and suggestions on our paper. We are looking forward to hearing from you soon. Best regards.